# How does the online innovation community climate affect the user's value co-creation behavior: The mediating role of motivation

Qiong Tan[1], Juan Tan[2], Xiaohui Gao[3]*

1 College of Economics, Central South University of Forestry and Technology, Changsha, China, 2 Business School, Beijing Technology and Business University, Beijing, China, 3 Business School, University of International Business and Economics, Beijing, China

* gxhiris@163.com

**Data Availability Statement:** All relevant data are within the manuscript and its Supporting Information files.

**Funding:** This manuscript was supported by the National Social Sciences Foundation of China

## Abstract

Online Innovation Community (OIC) serves as a virtual space for users to exchange products and services, and share knowledge and information. Previous studies have indicated that community climate is an important factor affecting users' value co-creation behavior, however, the influencing process has not been clearly revealed from the perspective of motivation. In this study, we explored the relationship between online innovation community climate (supportive climate and controlling climate), user motivation and value co-creation behavior (user's participation behavior and user's citizenship behavior) based on the SOR model. The study sample included 29,835 pieces of information from 3,315 users in 14 product sections of the OnePlus Community which were analyzed with Mplus8.1. The findings revealed that: (1) The supportive climate had a positive impact on user's citizenship behavior(β = 0.042), while the controlling climate exerted a significant positive impact on user's citizenship behavior (β = 0.078) and user's participation behavior(β = 0.099); (2) The need for achievement played a suppressing effect between community climate and user's participation behavior, the need for power played a suppressing effect between supportive climate and user's value co-creation behavior, and the need for affiliation played a mediating role between supportive climate and user's citizenship behavior (β = 0.010) and user's participation behavior(β = 0.006); (3) Community trust positively moderated the relationship between the need for achievement and user's participation behavior(β = 0.058) as well as between the need for power and user's participation behavior(β = 0.043).

## Introduction

Online Innovation Community (OIC) is a virtual community that provides users with feedback and enables exchanges on products or services [1]. It can effectively collect users' ideas to achieve effective integration of internal innovation and external ideas [2,3]. The development of digital technology and the digital economy has provided a new impetus for enterprises to continuously improve market competitiveness through OICs. Studies have shown that more

(Project No. 18CSH019), the Social Sciences Foundation of Beijing (Project No. 20JCC096), and Beijing Municipal Education Commission (Project No. SM201910011007). The funders had no role in study design, data collection and analysis, decision to publish, or preparation of the manuscript. We consider Dr Shihui Fu (Email: bringback@foxmail. com) and Dr Carla Ruiz-Mafe (Email: carla.ruiz@uv. es) are able to be potential reviewers of the manuscript.

**Competing interests:** The authors have declared that no competing interests exist.

than 80% of technology companies establish OICs to reduce R&D costs and promote product innovation, thereby improving corporate performance [4]. As a platform connecting enterprises and users, the OIC also serves as a social network for communication among users, which can bring substantial economic and market value to enterprises, and therefore has become an important platform for value co-creation between enterprises and users [5,6]. Earlier studies have demonstrated that user value co-creation is a process in which enterprises and users jointly create value and that it is a key factor affecting community development [7]. However, in practice, the user's value co-creation in most OICs has not achieved the desired results. For example, the decreasing number of active users and exhausted ideas in the DK community eventually led to its failure. This is attributed to two aspects. On the one hand, the short-term endogenous power of user value co-creation [6] and the external support for the community environment are weak [8], thus leading to insufficient motivation for user's value co-creation. On the other hand, due to the continuous expansion of the user base and the diversification of user groups in the OIC in the long run, it is difficult for the community to maintain a long-term relationship with users, thereby resulting in substantial user loss and weak stickiness [9]. These factors are detrimental to users' sustainable value co-creation.

Earlier studies on the impact of community environment on the user's value co-creation behavior focus primarily on the incentive effect of technology and system design of the platform [9–11], but little attention is paid to the important role of community climate [6,8]. The impact of community climate on user's value co-creation behavior is manifested in two aspects. On the one hand, a climate that meets user needs can promote their value co-creation behavior, which is viewed as the result of co-evolution and continuous interaction between users and the community [9,12]. The active, interactive climate in the community provides conditions for community users to participate in discussions and information exchanges, which can affect their attitudes and behaviors [13]. On the other hand, the community climate can give full play to the rigidity of the formal organization and the alternative role of the system [14], thereby controlling the uncivilized behavior in the community, generating a good psychological experience among users, and stimulating them to participate in value co-creation. Since the OIC is an informal organization where users are anonymous and virtual [11], the community climate can make up for the absence of formal contracts to a certain extent and play a role in regulating users' behaviors. On this basis, this study explored the different impacts of a supportive climate and a controlling climate in the community on user's value co-creation behavior [15–17]. Meanwhile, the present study categorized user value co-creation behavior into user's participation behavior and user's citizenship behavior [12], attempting to explore the differences between these two different user behaviors in terms of driving factors and behavioral performances. Furthermore, motivation refers to the impetus for users to participate in various activities to meet their needs [16]. When perceiving different community climates, users will experience changes in internal states and psychology, which in turn will affect their subsequent behaviors. Therefore, this study attempted to examine the path of influence of community climate on users' value co-creation behavior through the mediating effect of motivation. Furthermore, according to the Stimulus-Organism-Response (SOR) model, the environment as a stimulus will affect people's internal state and then their external reaction [18]. Therefore, the formation of user's value co-creation behavior in OIC may be that different community climate promotes users' different motives in the community, and then stimulates their value co-creation behavior [19]. In addition, trust is the basis of communication and cooperation between users in online communities, which can deepen the identity between users, reconcile conflicts and promote the sharing of information and resources in the community [20]. Different levels of trust among community users can affect their value-co-creation behaviors. Therefore, the present study also considered the moderating effect of trust between users' motivation and value co-creation behavior.

Hence, based on the Stimulus-Organism-Response (SOR) model, the present study sampled product information and user information from 14 product sections in the OnePlus Community and tested the impact path of community climate in OICs on user's participation behavior and user's citizenship behavior, and analyzed the mediating role of user's motivation and the mediating role of community trust with Mplus8.1.

## Literature review

### OIC climate

Climate refers to the dynamic and complex relationship between environmental changes and human behaviors and is an individual's unique perception of the environment [21,22]. The existing research on community climate is mainly based on organizational climate. Organizational climate includes subjective and objective perspectives: from the perspective of individual perception, the former believes that organizational climate is the feeling of employees about their environment, which is the sum of psychological climate [23,24]; from the organizational perspective, the latter posits that organizational climate is an objective description of organizational attributes, reflecting certain characteristics or attributes unique to organizations [25]. The subjective perspective is dominant in earlier studies on community climate and has been widely used in many fields, such as sociology, marketing, and organizational innovation. The online community climate in this study is also based on a subjective perspective.

Online community climate refers to the user's common perception of recurring attitudes, practices, and behavioral patterns in the community [26]. Since online communities, as an informal organization, are loose and open, community managers seldom intervene directly with users, but subtly influence their perceptions and behaviors by creating different community climates. Existing scholarship categorizes online community climate into humorous climate [27], social climate [28], team error management climate [29], virtual Brand Community's Innovation climate [30] etc. Some scholars believe that multiple climates co-exist in the same community, which are related to each other and work together, such as an innovative climate and a social climate [8], a supportive, fair and ethical climate [6], and an innovative, interactive and controlling climate [26], etc. In light of previous studies [15,16,29] and the practice of the community as the main object, the present study divided the OIC climate into a supportive climate and a controlling climate. A supportive climate refers to the climate in which users are encouraged to post freely, communicate on an equal footing, and develop friendly relationships in the community. On the contrary, a controlling climate means restricting the flow of information in the community and controlling the words, deeds and behaviors of members [29].

From the perspective of a supportive climate, the interaction and communication between users can enhance their sense of belonging to the community and thus encourage them to participate in community activities continuously [31]. For community managers, creating a supportive climate within the community is of great significance to realizing community empowerment, maintaining existing members, and attracting new members [17]. Generally, community prestige, home page display or community rewards are used to affirm user's participation behaviors, provide users with positive feedback, and make them feel the supportive climate provided by the community. In terms of controlling climate, by formulating certain systems and norms for the community, a code of action for community members is established, which can purify community information, maintain the normal operation of the OIC [32], and enhance the user's sense of identity and belonging to the community. Such a climate is embodied in the control of user identity and uncivilized behavior [16]. Zhu's (2020) research shows that supportive and controlling climate can significantly enhance users' innovative behavior [33].

## Value co-creation

The theory of value co-creation originates from the co-production of value. As a new value-creation model, value co-creation refers to a process in which enterprises, users and other stakeholders join forces to create value [34]. Current research on value co-creation mainly includes the following four branches: user experience-based value co-creation [35], service-oriented value co-creation [36], user-led value co-creation [36], and value co-creation based on user value originality [37]. Among them, the early service-oriented value co-creation focused primarily on the interaction between enterprises and users. With the advancement of science, technology and the digital economy, studies began to illuminate the interaction among more diverse subjects in the process of value creation. The research on service-oriented value co-creation is continuously upgraded and developed [38]. This study also examined service-oriented value co-creation. Here it is defined as the process of creating value through direct or indirect interaction between enterprises and users or between users and users by integrating the resources of both parties.

With the development of network technology, social interaction in online communities has become an important way for users to create value together [39]. Akman et al. believe that users can create value through information sharing and feedback interactions in online communities [1]. The value co-creation discussed in this study means that in the online innovation community, users directly communicate and interact with enterprises and other users, participate in new product development, provide feedback, etc., which helps enterprises to understand user needs and continuously improve products. Previous studies emphasize how to promote user participation in value co-creation in online communities, for example, how virtual community users create value through social network construction, impression management, community participation, and brand usage [40]. Users' helping behaviors, suggestions and rapport, feedback, and information contributions serve the purpose of value co-creation [1]. In addition, some scholars pay attention to the ubiquitous value co-creation behavior of users. For instance, Yi and Gong divided the user's value co-creation behavior into user's participation behavior and user's citizenship behavior, which was followed in this study [12]. User's participation behavior refers to the active participation of users in value-creating activities, such as information search, information sharing, responsible behavior, and interpersonal interaction. User's citizenship behavior refers to the behavior of users to maintain the interests of community users and create additional value spontaneously in the community, such as feedback, help, support and tolerance.

Furthermore, researchers have explored the influencing factors of value co-creation extensively and deeply from two aspects. On the one hand, from the perspective of users, studies have paid attention to their motivation and behaviors. For example, Lan et al. pointed out that self-efficacy, responsibility perception, reward expectation and learning process can affect users' value co-creation behavior [41]. Liu and Li indicated that users' interaction behaviors, including help-seeking, interaction and social behaviors, have a significant impact on their value co-creation behaviors [42]. Bui & Jeng pointed that user's sense of belonging can affect user's citizenship behavior [43]. On the other hand, from the perspective of communities, researchers have examined their governance mechanisms, incentives, and climates. For example, Wu et al. believe that the governance mechanism of the community affects the user's psychological ownership in the community and then impacts their value co-creation behaviors [9]. Ind et al. suggest that community incentives and community culture can promote user value co-creation [44]. Zhao et al. (2019) indicate that the supportive, fair and ethical climate of online communities can affect users' psychological ownership and their value co-creation

behavior [6]. Zhao et al. believe that the climate and trust of online communities will affect the willingness of users to create value together [45].

These studies show that both online community climate and user motivation are important factors influencing users' value co-creation behavior, and the inquiry into these two factors has yielded rich results. However, the mechanism whereby the two factors interact with each other in value co-creation remains unclear. At the same time, the impact of online community climate on users' ubiquitous value co-creation behaviors is under-researched. Therefore, this study analyzed the impact mechanism of user's participation behavior and user's citizenship behavior, respectively.

## Research hypothesis

Based on the SOR model, this study explored the relationship between the OIC climate and the user's value co-creation, as well as the internal transformation path of the user's value co-creation behavior. Specifically, the environment as an external stimulus affects the user's psychological state and their behavioral responses. Stimulus (S) represents environmental factors, which can cause changes in the user's psychological state. Different community climates can lead to different psychological changes of users in the community. Organism (O) is the user's internal emotional state, reflecting the psychological change of the user in the interaction process between the environment and the individual. Response (R) is the final behavior result of the user. In addition, many studies have revealed that the value co-creation behavior of users in the community is affected by community trust. As such, this study introduced community trust into the research model to explore its possible mediating role between user motivation and value co-creation behavior. We thus proposed the conceptual model of this study as shown in Fig 1.

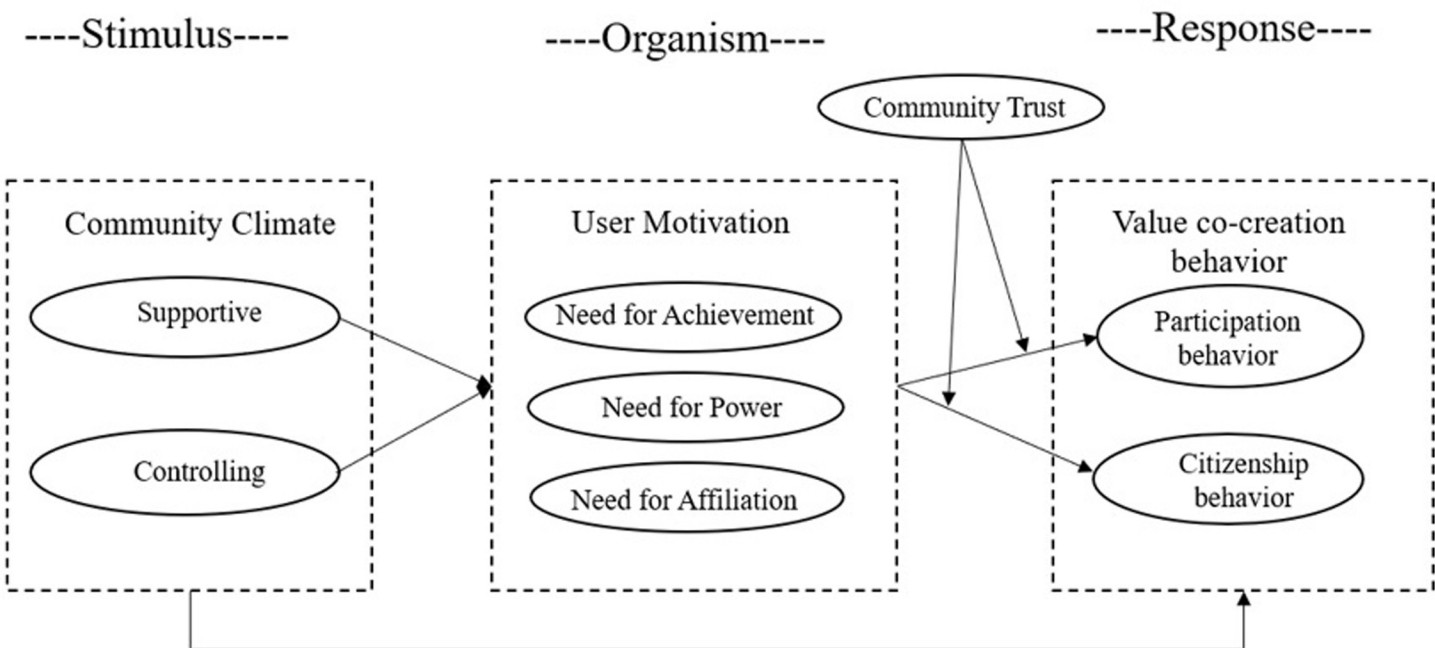

**Fig 1. Conceptual model.**

## Community climate and user's value co-creation

A supportive climate encourages free and equal communication and active interaction between users in the community. For this reason, community managers will promote user participation by rewarding active users, pinning and promoting high-quality content posted by users. This supportive climate can provide supportive resources and help for the development of users, thus satisfying users' creation needs and stimulating their participation motivation [46,47]. The empirical research of Zhao and Jing et al. showed that the supportive climate in the online brand community could strengthen the motivation of users to participate in the community as well as their innovation [16]. Cao et al. believed that a supportive climate in the community could promote user participation in the community [48]. When they feel the need for respect, recognition and belonging in the community, or get help and responses from other members, this will significantly enhance their participation in the community. Zhao et al. indicated that when community users perceive the supportive climate of the community, their value co-creation behavior will be remarkably improved [6]. In this study, we argue that community users' perception of the online community's supportive climate positively affects their value co-creation behavior. When users perceive a strong supportive climate in the community, they are more likely to actively engage in value co-creation. Therefore, we proposed the following hypotheses:

H1a: The supportive climate positively affects user's participation behavior.

H1b: The supportive climate positively affects user's citizenship behavior.

The controlling climate is the management and regulation of users by the community. By building a code of conduct for community users, this climate can purify community information and content, and rectify users' behaviors that do not conform to community norms. As a result, users can focus on information exchange and resource acquisition in the community, enhance the identity of the community and promote their participation behavior. Research by Zhang et al. exhibited that the management and control of communities can guide community users to regulate their behavior and enhance their community identity [49], and community identity will effectively promote users' citizenship behavior [50]. Zhao and Jing et al. believed that managing and controlling the online community environment could improve the enthusiasm of community users [16], thereby encouraging them to produce innovative behaviors. Moreover, users in online communities generally have diverse backgrounds and knowledge. In the absence of proper guidance, users' willingness to participate in community activities may be reduced, which will hinder the development of communities [26]. The controlling climate in the community can maintain its normal operation, put normative pressure on users, enable them to focus on community activities, and promote their participation in the community. These studies have established that the controlling climate can purify the community environment, stimulate community users' willingness to participate in activities in a good community environment, and promote value co-creation behavior by effectively managing community users. Therefore, we proposed the following hypotheses:

H2a: The controlling climate positively affects user's participation behavior.

H2b: The controlling climate positively affects user's citizenship behavior.

## The mediating role of motivation

Motivation originates from internal needs and is the impetus for users to participate in various activities to satisfy their internal needs [19,44]. The need for achievement theory holds that users hope to successfully complete work goals through their own efforts or try their best to do

their work better. Based on their internal needs, users usually set higher goals, are willing to face competition, expect to surpass others, challenge themselves and pursue innovation. McClelland proposed that many human needs are social [51]. He believed that new needs would arise when the survival needs are satisfied, and classified them into three categories: Need for Achievement, Need for Affiliation and Need for Power. The online community is essentially a social network based on interpersonal relationships whereby users communicate with each other, share knowledge and information, and thus gain social identity. Therefore, users participate in activities in the community and cooperate with other users primarily because of the sense of accomplishment obtained by exerting their abilities, the sense of belonging obtained through the establishment and maintenance of interpersonal relationships, and the establishment of personal influence in the community. In a word, they are driven by motivations such as achievement, affiliation, and power.

The need for achievement of OIC users refers to the need of users to achieve goals and pursue excellence based on their own interests and passions [52]. According to earlier studies, users' need for achievement can affect their enthusiasm to participate in community activities and their participation behavior. Zhao's research confirmed that users' need for achievement could significantly affect their participation behavior [53]. Users with higher needs for achievement realize their intrinsic needs by continuously solving the problems of other users in the community [54]. When feeling a free or fair community environment, users will devote themselves to community activities, thus gaining a sense of accomplishment and satisfaction from successful self-challenge [16,55]. Therefore, we proposed the following hypotheses:

H3a: The need for achievement plays a mediating role between the supportive climate and the user's value citizenship behavior.

H3b: The need for achievement plays a mediating role between the supportive climate and user's participation behavior.

H3c: The need for achievement plays a mediating role between the controlling climate and user's citizenship behavior.

H3d: The need for achievement plays a mediating role between the controlling climate and user's participation behavior.

The need for power refers to the user's need to influence or control others by acquiring social status [52]. Users influence others by providing valuable information, actively participating in community activities, earning the respect of other users, and gaining community status. The stronger the need for power is, the stronger the desire to influence other people's thoughts or behaviors is. In virtual communities, the behavior of influencing others is mainly manifested in actively participating in community management and strengthening communication with other users. When users feel that they have different permissions from other users in the community, their need for power is met. To continue this privilege and gain more power, they will actively participate in value co-creation in the community. Therefore, we proposed the following hypotheses:

H4a: The need for power plays a mediating role between the supportive climate and user's citizenship behavior.

H4b: The need for power plays a mediating role between the supportive climate and user's participation behavior.

H4c: The need for power plays a mediating role between the controlling climate and user's citizenship behavior.

H4d: The need for power plays a mediating role between the controlling climate and user's participation behavior.

The need for affiliation refers to the need of users to pursue friendliness and value interpersonal relationships [52]. Community managers encourage users to freely exchange information in the community and actively participate in interactions, but also require them to post in a civilized manner and control uncivilized behaviors in the community so that users feel respected in the community. In this way, they will feel that they have established connections with others and are more willing to pay attention to others in communication. According to the social network incentive theory, the social network of OIC indicates that members are not independent individuals but social subjects influenced by social relationships with obvious social attributes [42]. Users share knowledge, learn from each other, exchange technology, and express opinions and thoughts in the social network so as to obtain emotional and information satisfaction. Users with higher needs for affiliation are more willing to participate in value co-creation activities in the community in order to maintain a friendship with others and gain recognition and love from others. Therefore, we proposed the following hypotheses:

H5a: The need for affiliation plays a mediating role between the supportive climate and user's value citizenship behavior.

H5b: The need for affiliation plays a mediating role between the supportive climate and user's participation behavior.

H5c: The need for affiliation plays a mediating role between the controlling climate and user's citizenship behavior.

H5d: The need for affiliation plays a mediating role between the controlling climate and user's participation behavior.

## The moderating role of community trust

Trust is a crucial factor that affects users' participation and knowledge sharing in the community. The level of users' trust in the community will affect their behaviors in the community [56]. At different levels of trust, motivation will have different effects on user behavior [57]. Community trust means the user's trust in the community climate and environment. When users have a high degree of trust in the community, users will actively participate in community activities and share their own information, no matter whether it is driven by the needs of other members or the motivation of users to satisfy their self-achievement [49]. Zhang and Liu indicated that users with a higher degree of trust in the community perceive higher levels of information support and risks from the community and thus have more trust in the community [48], which can generate a virtuous circle. In the present study, therefore, we argue that when users have the motivation to satisfy their need for achievement in the community, those with higher trust in the community will be more willing to participate in community interaction than users with lower trust in the community. Therefore, we proposed the following hypotheses:

H6a-H6b: Community trust plays a positive moderating role between the need for achievement and user's participation/citizenship behavior.

H6c-H6d: Community trust plays a positive moderating role between the need for power and user's participation/citizenship behavior.

H6e-H6f: Community trust plays a positive moderating role between the need for affiliation and user's participation/citizenship behavior.

## Research design

### Sampling and data collection

This study analyzed the OnePlus Community (https://www.oneplusbbs.com), established in 2013 by OnePlus mobile phone officials. It is a virtual community where users can exchange products and services, as well as share knowledge and information. In OnePlus community, users have the opportunities to express their views and share their experiences. According to the report by Counterpoint in the first half of 2023, OnePlus was recognized as one of the top ten brands with the fastest growth in global market share and it had more than 15 million community users in 196 countries around the world. The users in OnePlus Community are highly active, the rich user resources owned by the community contribute to this research work.

In order to collect user data, members of the research group registered as community users and gathered information from the OnePlus community forum, including the number of new posts, the number of comments, and the number of posts selected as "hot". As required by the community, users can only discuss product experience and improvement suggestions in the corresponding product section. To collect users' product experience properly, we sampled 14 product sections in the community by Python (as of data collection, a total of 17 sections, excluding three sections beyond the research purpose, including Youleyuan, Qingsheying and Official activities) as the primary source for information collection. Since only the latest 1,000 pages of posts and related information (19 posts per page, 266,000 posts in total) from the time of data capture were available on the website of the target community, we were unable to obtain all the posts of the community since its establishment. Still, the crawled data were sufficient to meet our research needs, and the data posted up to October 4, 2021 were collected.

The data required for the research involves both users and products, which were collected in three rounds: in the first round, posts that were officially rated as "hot" in the community were collected, as shown in Fig 2, including the popularity value, product section where the post was located, and the link to the personal website. A total of 38,894 posts were collected. In the second round, leveraging the uniqueness of each user's personal website has been obtained in the first round, basic user information was collected, including user's name, ID, number of cheers, displayed items of user information, online time, number of friends, etc. In the third round, the number of comments deleted by the authority, the number of comments on the post, and the number of views on the post collected in the first round were collected. Finally, a total of 676,380 pieces of information were collected.

The inclusion and exclusion criteria are as follows: (1) official activity posts; (2) posts and user information with incomplete information; and (3) only one post randomly included for the same user. As the collection of user information data in this study was related to the posts by users, if all posts by the same user were included, the same user information would be collected repeatedly for users who post multiple times, which would cause problems for data cleaning and sorting. In addition, we randomly deleted multiple posts from the same user to ensure the accuracy of the results. Totally 29,835 pieces of information from 3,315 users were finally included for analysis.

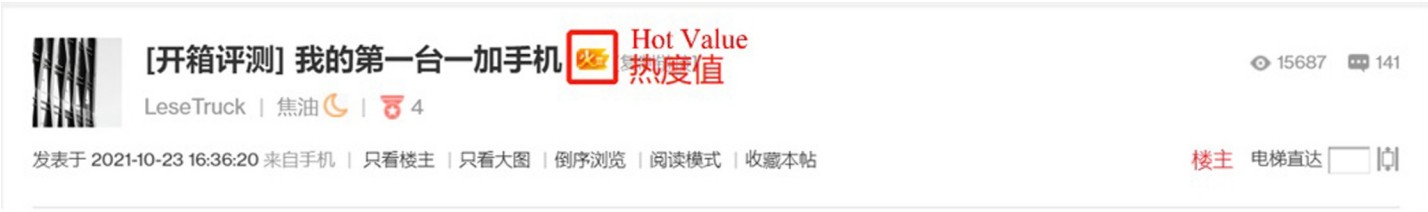

**Fig 2. "Hot" posts in the OnePlus community.**

## Variables

Each variable was defined and measured with reference to earlier studies and the context of the OnePlus Community, as shown in Table 1.

(1) Dependent variable: User participation behavior (UPB) refers to the behavior that users actively participate in value-creating activities in the community, so it was measured by the number of posts by users[12]; user citizenship behavior (UCB) refers to the behavior that users spontaneously safeguard users' interests and create additional value in the community, which is measured by the number of comments posted by users [12].

(2) Independent variable: Supportive climate (SC) refers to the climate in which users are encouraged to post freely, communicate on an equal footing and develop friendly relations [16], so it was measured by the hot value of posts marked by the community. Controlling climate (CC) refers to the flow of information in the community, the words, deeds and behaviors of members were restricted [16], so it was measured by the number of comments on posts that were officially deleted.

(3) Mediating variable: The need for achievement (NA) refers to the need of users to achieve goals and pursue excellence based on their own interests and passions [43], so it was represented by the number of cheers users get in the OnePlus Community [56]. The need for power (NP) refers to the user's need to influence or control others by acquiring social status [56], so it was indicated by the different user group levels that users belong to in the community. There are 11 user groups in the OnePlus Community, and different user group levels are set according to the behaviors of users participating in community activities. Users with stronger community power have a wider scope of activities and comment permissions than users with weaker power. The higher the user group level is, the stronger the influence on others is. The need for affiliation (NFA) refers to the need of users to pursue friendliness and value interpersonal relationships [56], so it was measured by the number of friends a user has in the community.

(4) Moderating variable: Community trust (TR) refers to the user's trust in the community [52]. In this study, we argue that users think they belong to a certain group. The higher the identification with the community is, the more they believe that the community can represent their own image, the more they trust the community, the more likely they are to display their personal information in the community. Therefore, community trust was measured by items in which users display personal information in the community.

**Table 1. The description and definition of each variable.**

| Type | Variable | Symbol | Description | Reference |
|---|---|---|---|---|
| Dependent | User's Participation behavior | UPB | The number of posts by user $i$ | Yi and Gong (2013) [12] |
| | User's Citizenship behavior | UCB | The number of comments by user $i$ | |
| Independent | Supportive climate | SC | The hot value of the post | Zhao and Jing (2016) [16] |
| | Controlling climate | CC | The number of deleted comments as shown in the post | Zhao et al. (2019) [6] |
| Mediating | Need for achievement | NA | The number of cheers for user i in the community | Wann and Sukoco (2010) [52] |
| | Need for power | NP | The level of the user group to which user $i$ belongs | Thompson et al. (2008) [56] |
| | Need for affiliation | NFA | The number of friends of user $i$ in the community | |
| Moderating | Community trust | TR | The display items of user $i$'s information | Wann and Sukoco (2010) [52] |
| Control | User age in community | UAC | The interval from user registration to data collection | Tan et al. (2022) [58] |
| | Product section in community | PSC | The product section to which the post belongs | |

(5) Control variable: User age in community (UAC): UAC can have a significant impact on users' participation in community activities [58]. In this study, it was represented by the time between user registration and data collection. Product section in community (PSC): the OnePlus Community has set up multiple product sections according to product types and stipulates that users must post content under the corresponding section; otherwise, the post will be deleted. For users, they often have rich experience in using the products they purchased, so they are more likely to share their experience in a fixed product section instead of participating in the activities of other product sections. For this reason, we used PSC as a control variable, which was indicated by the product section to which the user's content belongs.

## Results

### Descriptive statistical analysis

The descriptive statistical analysis results are shown in Table 2. First, the mean of user comments was 456.70, and the average number of active posts was 24, with a significant difference between them. Secondly, the average value of "hot posts" marked by the community was 121.93, and the average number of deleted comments of users in the community was 2.34, indicating that users can perceive a strong supportive climate, and most of them can abide by community norms. In addition, due to the remarkable differences in variables such as the need for achievement, user's participation behavior, and user's citizenship behavior, we conducted Ln(x+1) processing on some variables. After logarithmization, related descriptive statistics and correlation analysis are shown in Table 3.

### Correlation analysis

It can be seen from Table 3 that there was a correlation between the main variables, and the maximum correlation coefficient between the variables was 0.648. The maximum variance inflation factor (VIF) among the variables was 1.791, with a mean value of 1.30, which was far less than 10, indicating that the multicollinearity was within the acceptable range.

### Hypothesis testing

**Structural equation model.**   Mplus8.1 was used for fitting data of the model. The fitting results were CFI = 0.988, TLI = 0.923, REMSA = 0.069, SRMR = 0.028, all within the threshold

**Table 2. Descriptive statistical analysis.**

| Variable | Sample size | Mean | SD | Min. | Max. |
|---|---|---|---|---|---|
| SC | 3315 | 121.93 | 572.50 | 0 | 13237 |
| CC | 3315 | 2.34 | 7.54 | 0 | 160 |
| NA | 3315 | 9.96 | 40.98 | 0 | 1331 |
| NP | 3315 | 7.09 | 1.89 | 1 | 11 |
| NFA | 3315 | 11.8 | 38.74 | 0 | 645 |
| TR | 3315 | 1.20 | 1.85 | 0 | 14 |
| UPB | 3315 | 24 | 47.36 | 0 | 2104 |
| UCB | 3315 | 456.70 | 861.89 | 0 | 33291 |
| PSC | 3315 | — | 2.64 | 1 | 10 |
| UAC | 3315 | 4.76 | 1.92 | 0.28 | 8 |

**Table 3. Correlation coefficients.**

| Variable | Ln SC | Ln CC | Ln NA | NP | Ln NFA | TR | Ln UPB | Ln UCB |
|---|---|---|---|---|---|---|---|---|
| 1 | 1.791 | | | | | | | |
| 2 | 0.626*** | 1.667 | | | | | | |
| 3 | 0.411*** | 0.335*** | 1.227 | | | | | |
| 4 | -0.077*** | -0.030 | -0.047*** | 1.022 | | | | |
| 5 | 0.075*** | 0.067*** | 0.050*** | 0.113*** | 1.049 | | | |
| 6 | 0.016 | 0.024 | 0.082*** | 0.036*** | 0.168 | 1.036 | | |
| 7 | -0.022 | 0.047*** | -0.106*** | 0.426*** | 0.178*** | 0.032 | — | |
| 8 | 0.064*** | 0.104*** | 0.046*** | 0.648*** | 0.266*** | 0.122*** | 0.528*** | — |
| Mean | 0.68 | 0.24 | 0.59 | 7.09 | 0.18 | 1.20 | 1.12 | 2.39 |
| SD | 1.05 | 0.35 | 0.46 | 1.89 | 1.78 | 1.85 | 0.47 | 0.51 |

\* $p < 0.05$

\*\* $p < 0.01$

\*\*\* $p < 0.001$.

The value on the diagonal is the VIF of the corresponding variable.

standard. The structural equation model is shown in Fig 3, and the path coefficient test results are shown in Table 4.

As shown in the path coefficient test results, the standard errors (S.E.) were all within the theoretical range, without extreme values, and met the requirements. The results showed that the supportive climate had no significant impact on user's participation behavior (β = -0.012, $p > 0.05$). H1a was not validated and had a significant effect on user's citizen behavior (β = 0.042, $p < 0.001$), and H1b was validated. The controlling climate had a remarkable impact on user's participation behavior (β = 0.099, $p < 0.001$) and user's citizenship behavior (β = 0.078, $p < 0.001$), and H2a and H2b were validated. In addition, this study tested the impact of community climate on user motivation, and the results showed that except for the $p$-values of controlling climate as well as the need for power and affiliation, the rest of the $p$-values were all less than 0.05, indicating that the two were significantly correlated. We also tested the impact of motivation on their value co-creation behavior, and the results demonstrated that except for the $p$-values of the need for achievement and user's citizenship behavior, the rest of the $p$-values were all less than 0.05, suggesting that the two were significantly correlated.

**The mediating role of the user motivation.** Mplus8.1 and the Bootstrap method (sample size: 5,000, CI: 95%) were used to estimate and test the multiple mediating roles of the need for achievement between community climate and user's value co-creation behavior. The significance of the mediating effect was determined based on whether the confidence interval contained 0, as shown in Table 5.

The confidence interval for the mediating role of the need for achievement between community climate (supportive and controlling) and user's citizenship behavior contained 0, indicating that there was no mediating effect between the need for achievement and user's citizenship behavior, and H3a and H3c were not validated. There is a negative significant influence of supportive climate (coefficient c) on user's participation behavior. The supportive climate has a positive impact on the need for achievement, and the interval does not contain 0 at (LLCI = 0.286, ULCI = 0.375). Additionally, it was found that the need for achievement has a significant negative effect on user's participation behavior with an interval that does not contain 0 at (LLCI = -0.159, ULCI = -0.087). Furthermore, after controlling for the need for achievement, it was observed that supportive climate did not have any significant influence on

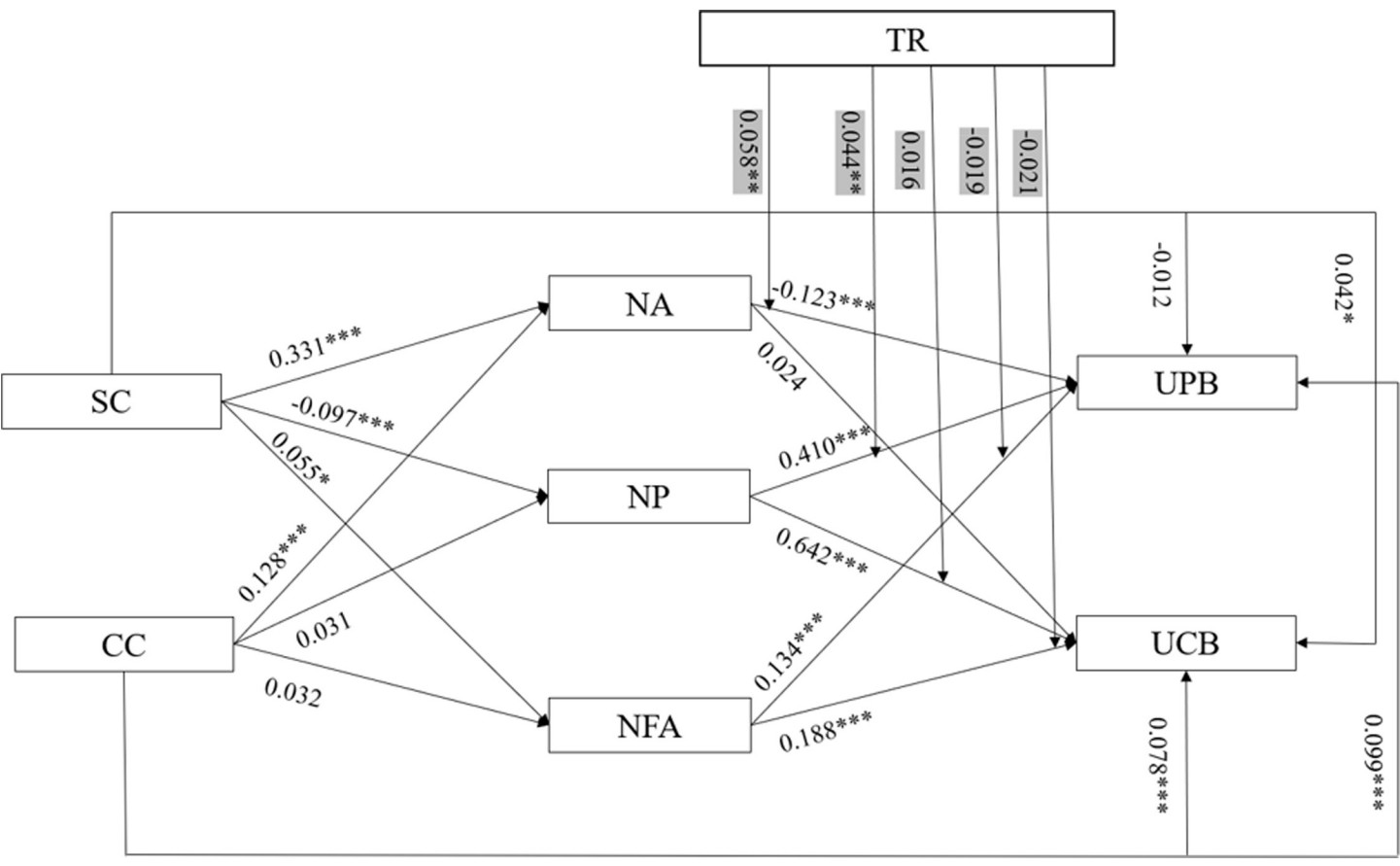

**Fig 3. Structural equation model with moderation results.** * $p < 0.05$, ** $p < 0.01$, *** $p < 0.001$.

users' participation behavior. Therefore, the need for achievement plays a negative mediating role between supportive climate and user's participation behavior; H3b has not been verified. The influence of controlling climate on user's participation behavior (coefficient c) is not significant. There is a positive impact of controlling climate on the need for achievement (coefficient a), and this interval does not include 0 at (LLCI = 0.081, ULCI = 0.177). The need for achievement has a significant negative effect on user's participation behavior with an interval that does not contain 0 at (LLCI = -0.159, ULCI = -0.087). After controlling for the need for achievement in this relationship between controlling climate and user's participation behavior (c'), it was found to be positively significant. The need for achievement has a suppressing effect in the relationship between controlling climate and user's participation behavior [49]. It was also found that a*b is -0.0157, which accounts for |ab/c'| = |-0.0157/0.099| = 15.86% of the direct effect. H3d has not been verified.

There is a negative significant influence of supportive climate (coefficient c) on user's participation behavior. The influence of supportive climate on the need for power is also negatively significant with an interval that does not include 0 at (LLCI = -0.142, ULCI = -0.051). Moreover, the need for power has a positive effect on user's participation behavior and this interval does not contain 0 at (LLCI = 0.381, ULCI = 0.439). After controlling for the need for power, the relationship between supportive climate and user's participation behavior is not significant. Therefore, the need for power plays a negative mediating role between supportive climate and user participation behavior, H4b has not been verified. The influence of supportive

**Table 4. Path analysis and test results of the research model.**

| Path | Estimate | S.E. | Est./S.E. | P value |
|---|---|---|---|---|
| SC-UPB | -0.012 | 0.021 | -0.557 | 0.578 |
| SC-UCB | 0.042 | 0.017 | 2.423 | * |
| CC-UPB | 0.099 | 0.021 | 4.758 | *** |
| CC-UCB | 0.078 | 0.017 | 4.719 | *** |
| SC-NA | 0.331 | 0.023 | 14.671 | *** |
| SC-NP | -0.097 | 0.023 | -4.235 | *** |
| SC-NFA | 0.055 | 0.023 | 2.431 | * |
| CC-NA | 0.128 | 0.025 | 5.130 | *** |
| CC-NP | 0.031 | 0.023 | 1.359 | 0.174 |
| CC-NFA | 0.032 | 0.022 | 1.440 | 0.150 |
| NA-UPB | -0.123 | 0.018 | -6.742 | *** |
| NA-UCB | 0.024 | 0.015 | 1.585 | 0.113 |
| NP-UPB | 0.410 | 0.015 | 27.732 | *** |
| NP-UCB | 0.642 | 0.012 | 54.982 | *** |
| NFA-UPB | 0.134 | 0.016 | 8.559 | *** |
| NFA-UCB | 0.188 | 0.013 | 14.934 | *** |

\* $p < 0.05$

\*\* $p < 0.01$

\*\*\* $p < 0.001$.

climate on users' citizenship behavior (coefficient c) is negatively significant, the influence of supportive climate on the need for power (coefficient a) is negatively significant, and the interval (LLCI = -0.142, ULCI = -0.051) does not include 0; the influence of the need for power on user's citizenship behavior (coefficient b) is positive, and the interval (LLCI = 0.619,

**Table 5. Bootstrap analysis of the mediating effect.**

| Path | Estimate | P value | 95% CI | | | |
|---|---|---|---|---|---|---|
| | | | Uncorrected | | Corrected | |
| | | | Lower limit | Higher limit | Lower limit | Higher limit |
| SC-NA-UPB | -0.041 | *** | -0.023 | -0.013 | -0.051 | -0.030 |
| SC-NP-UPB | -0.040 | *** | -0.025 | -0.011 | -0.055 | -0.024 |
| SC-NFA-UPB | 0.007 | * | 0.001 | 0.006 | 0.002 | 0.013 |
| CC-NA-UPB | -0.016 | *** | -0.030 | -0.013 | -0.023 | -0.010 |
| CC-NP-UPB | 0.013 | 0.176 | -0.004 | 0.037 | -0.003 | 0.028 |
| CC-NFA-UPB | 0.004 | 0.159 | -0.001 | 0.013 | -0.002 | 0.009 |
| SC-NA-UCB | 0.008 | 0.113 | 0.000 | 0.008 | -0.002 | 0.016 |
| SC-NP-UCB | -0.062 | *** | -0.042 | -0.018 | -0.087 | -0.038 |
| SC-NFA-UCB | 0.010 | * | 0.002 | 0.008 | 0.003 | 0.017 |
| CC-NA-UCB | 0.003 | 0.139 | 0.000 | 0.010 | 0.000 | 0.007 |
| CC-NP-UCB | 0.020 | 0.174 | -0.006 | 0.063 | -0.008 | 0.044 |
| CC-NFA-UCB | 0.006 | 0.154 | 0.073 | 0.152 | -0.001 | 0.013 |

\* $p < 0.05$

\*\* $p < 0.01$

\*\*\* $p < 0.001$.

ULCI = 0.664) does not include 0; Controlling the need for power, the influence of controlling climate on user's citizenship behavior (coefficient c') is positively significant. Coefficient ab and c' are different signs. Therefore, the need for power has a suppressing effect in the relationship between supportive climate and user's citizenship behavior [59], a*b is -0.062, accounting for |ab/c'| = |-0.062/0.042| = 147.6% of the direct effect, and H4a has not been verified. The interval of indirect effect between controlling climate and user's value co-creation behavior (citizenship behavior and participation behavior) contains 0, which indicates that there is no mediating effect between supporting climate and user's value co-creation behavior, and H4c and H4d have not been verified.

The confidence interval for the indirect effect of the need for affiliation between supportive climate and user's value co-creation behavior (citizenship behavior and participation behavior) did not contain 0, and that between controlling climate and user's value co-creation behavior (citizenship behavior and participation behavior) contained 0. This indicated that the need for affiliation had a mediating effect between supportive climate and user's value co-creation behavior, but it had no mediating effect between controlling climate and user's value co-creation behavior; therefore, H5a and H5b were validated, while H5c and H5d were not validated.

It is worth noting that the need for achievement played a negative mediating role between supportive climate and user's participation behavior, and played a suppressing effect between controlling climate and user's participation behavior. A similar result can be drawn back to the original data we collected while considering the reality of the OnePlus Community: the vast majority of users got more cheers from comments than posts, which means that user's citizenship behavior had a greater impact on the need for achievement than user's participation behavior, which also responds to the relevant conclusions of other scholars [60]. If the main function of the OnePlus Community is regarded as a virtual brand community, then its value co-creation behavior is mainly citizenship behavior. Thus, it is not difficult to understand that users with higher needs for achievement will reduce participation behavior and be more obsessed with increasing their citizenship behavior.

The need for power played a negative mediating role between supportive climate and user's participation behavior, and played a suppressing effect between supportive climate and user's citizenship behavior. According to the social power theory proposed by French and Raven, the five sources of social power are reward, coercion, legitimacy, expertise and referent [61]. But users gain power in the community mainly through expertise and referential power. Expertise is mainly demonstrated through posts and comments, and referential rights are mainly obtained through positive feedback from other users. However, the purpose of creating a supportive community climate is more focused on how to encourage ordinary users to actively participate in community value co-creation activities. In the empirical analysis, we used the number of posts with a high number of replies and views within a certain period of time, that is, the number of "hot posts" to measure the supportive climate of the community, which essentially reflects the user's behavior. The attention theory states that in a world of noisy or abundant information, attention is focused on some aspects and neglects others [62]. If OIC users are eager to make "hot posts," this will attenuate their pursuit of community power. In the empirical study, if users allocated their attention to the number of replies and views of a certain post, they would reduce the frequency and quality of posting as well as the enthusiasm to comment on other users' posts. The former reflected the supportive climate of the community, while the latter was a manifestation of the user's expertise and referential rights. Thus, we could understand the negative mediating role of the need for power between supportive climate and user value co-creation behavior.

### The moderating role of community trust

We employed hierarchical regression to test the moderating effect of community trust. Since the need for achievement did not have a significant impact on user's citizenship behavior, only the mediating role of community trust between the need for achievement and user's participation behavior, and the need for power and affiliation on user's value co-creation was analyzed. To avoid collinearity, the variables of the need for power and community trust were centralized before testing the mediating effect, and then the interaction item was calculated. Given the length constraints of the article, the valid results of the hypotheses were presented in Table 6.

In Model 3 of Table 6, it can be seen that the interaction term between the need for achievement and community trust had a positive impact on user participation behavior (β = 0.058, p < 0.01), thereby validating H6a. Additionally, the interaction term between the need for power and community trust exhibited a positive impact on user participation behavior (β = 0.044, p < 0.01), confirming H6b. However, other hypotheses were not validated.

## Conclusion and implications

### Conclusion and discussion

How to promote users' participation in value co-creation in online communities is an increasingly important topic. Based on the SOR model, this study sample included 29,835 pieces of information from 3,315 users in 14 product sections of the OnePlus Community which were analyzed with Mplus8.1, explore the impact of the online community climate on users' value co-creation behavior in the OnePlus Community and analyzed the mediating role of motivation and the moderating role of community trust. The findings are as follows:

(1) Supportive climate positively affected user's citizenship behavior, and controlling climate positively affected user's participation behavior and user's citizenship behavior. The result is in line with the findings of Zhao et al. [6], whose study showed that supportive climate can promote users' value co-creation behavior by enhancing their psychological capital. Zhu's (2020) [33] study showed that controlling climate can promote users' innovation behavior. The stronger the supportive and controlling climate provided by the community, the more it can promote users' information exchange and resource acquisition behavior in the community, thus promoting their value co-creation behavior in the community.

**Table 6. The test results of the moderating effect.**

| Variable | Participation behavior | | | Participation behavior | | |
| --- | --- | --- | --- | --- | --- | --- |
| | Model 1 | Model 2 | Model 3 | Model 4 | Model 5 | Model 6 |
| UAC | 0.220*** | 0.260*** | 0.264*** | 0.220*** | 0.011 | 0.012 |
| PSC | -0.015 | 0.015 | 0.024 | -0.015 | -0.027 | -0.027 |
| NA | | -0.128*** | -0.142*** | | | |
| NP | | | | | 0.413*** | 0.417*** |
| TR | | 0.143*** | 0.125*** | | 0.062*** | 0.056** |
| Interaction item | | | 0.058** | | | 0.044** |
| F | 94.066 | 39.087 | 9.355 | 94.066 | 267.109 | 7.510 |
| $R^2$ | 0.054 | 0.076 | 0.078 | 0.054 | 0.185 | 0.186 |
| $\Delta R^2$ | 0.054 | 0.022 | 0.003 | 0.054 | 0.131 | 0.002 |

\* $p < 0.05$

\*\* $p < 0.01$

\*\*\* $p < 0.001$.

(2) The need for achievement played a negative mediating role between supportive climate and user's participation behavior, the need for achievement played a suppressing effect between controlling climate and user's participation behavior; the need for power played a negative mediating role in supportive climate and user's participation behavior, the need for power played a suppressing effect in supportive climate and user's citizenship behavior; and the need for affiliation played an mediating role between supportive climate and user's value co-creation behavior, and similar findings were found in Wann-Yih & Sukoco's [52] study. These results showed that these three intrinsic motivations can promote the user's knowledge sharing behavior in OIC.

(3) Community trust had a positive moderating effect between the need for achievement and user's citizenship behavior, as well as between the need for power and user's citizenship behavior. The result is in line with the findings of Zhao and Detlor's (2021) [57] study, indicating that trust plays a moderating effect between the need for power and user's sharing intensions. This shows that the higher level of community trust strengthens the influence of users' motivation of achievement needs and power needs on their citizenship behavior.

## Theoretical contribution

The main theoretical contributions of this study are as follows:

(1) This paper reveals the influence mechanism of online community climate on users' value co-creation behavior, and expands the related research on user behavior in online community. Earlier studies have shown that the online community climate can have an impact on users' value co-creation behavior [6], but there is a lack of detailed research on the relationship between the two. In this study, we categorized community climate into supporting climate and controlling climate, users' value co-creation behavior into user's citizenship behavior and user's participation behavior, then we explored the impact of online community climate on these two dimensions, thus promoting the research on ubiquitous value co-creation behavior.

(2) There is a huge body of literature on online community climate and user's value co-creation behavior [6,27], but the mechanism whereby the two interact with each other remains unclear. It is found that achievement need motivation plays a suppressing effect between controlling climate and user participation behavior, and power need motivation plays a suppressing effect between supporting climate and user citizenship behavior. Therefore, this study introduced the user's motivation as a mediating variable, focused on the relationship between online community climate, user's motivation and user's value co-creation behavior, and explored the mediating role of the need for achievement, hence enriching the related research on user motivation in online communities.

(3) Users' different level of trust in the community will affect their behaviors in the community. Taking community trust as the moderating variable, the present study explored the moderating effect of community trust between users' motivation and value co-creation behavior [20], which enriches the research on community trust.

## Implications

(1) Strengthen the dynamic management of the OIC and maintain the balance between "openness" and "control". On the one hand, online communities create a positive and supportive climate by putting high-quality posts at the top and giving them different popularity, which

makes users in the community more active. On the other hand, it is necessary to properly control the community by various measures, such as controlling community information exchange, strictly reviewing the content of user posts, and stimulating users' value co-creation behaviors. Community activities are consistent with the innovation goals of enterprises, thus bringing more valuable innovation results to leading enterprises.

(2) Stimulate the intrinsic motivation of users to participate in value co-creation in the community. The community can improve user loyalty by increasing the user level settings in the community to stimulate their needs for power; at the same time, it can encourage users to make friends in the community, share their views and suggestions with others, meet their needs for affiliation, thus enabling them to have a sense of belonging to the community. The community should get rid of the reward system. In order to get rewards, users are more inclined to engage in easy commenting behaviors [63] rather than actively posting, which is prone to "threadjacking" behaviors [64], making it difficult to manage the community. Through the external stimulation of creating different community climates, online communities can affect the internal motivation of users in the community and transform the external impetus into the internal motivation of personal value co-creation.

(3) Enhance the trust within the community and between community users. On the one hand, the community can encourage community users to speak freely and express some unique and insightful views by creating a good supportive climate, so as to enhance users' perception of the supportive climate in the community. On the other hand, the community can strengthen supervision and control by creating a controlling climate, protect users' personal information, reduce the uncertainty of users' perception of the community, and reduce the risk of participating in community activities to improve the level of community trust.

## Limitations

First, this study sampled the cross-sectional data of the OnePlus Community product sections for analysis, but the role of community climate may change with time. In the future, longitudinal panel data can be considered to investigate the dynamic changes in the impact of community climate on user behavior. Second, the generality of the conclusions in this study needs to be further verified. We only explored the data in the OnePlus Community and failed to widely adopt product information and user information of other similar industries. Different types of online communities provide different supportive climates and controlling climates, which will affect users' psychological perceptions of the community and have different impacts on users' value co-creation behaviors. Third, while the exploration of users' value co-creation behavior in online innovation communities from the perspective of the community is significant, many issues still need to be explored in depth. More complex models may be used for in-depth research in the future. In addition, this paper only considers the mediating role of achievement motivation and the moderating role of community trust. In the future, we can consider different mediating variables, such as perceived benefits and moderating variables, such as user value concept, and explore the influence mechanism of community climate on user value co-creation behavior.

## Supporting information

**S1 Data.**
(XLSX)

## Author Contributions

**Conceptualization:** Qiong Tan.

**Investigation:** Qiong Tan.

**Methodology:** Qiong Tan, Xiaohui Gao.

**Supervision:** Juan Tan.

**Writing – original draft:** Qiong Tan, Xiaohui Gao.

**Writing – review & editing:** Juan Tan.

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
