## [Decision Letter · Decision Letter 0]

11 Oct 2023

PONE-D-23-29718How does the online innovation community climate affect the user's value co-creation behavior: the mediating role of motivationPLOS ONE

Dear Dr. Gao,

Thank you for submitting your manuscript to PLOS ONE. After careful consideration, we feel that it has merit but does not fully meet PLOS ONE’s publication criteria as it currently stands. Therefore, we invite you to submit a revised version of the manuscript that addresses the points raised during the review process.

The author can choose not to cite the article recommended by Reviewer 1 but needs to add some literature to enrich this study.

We look forward to receiving your revised manuscript.

Kind regards,

Chunyu Zhang

Academic Editor

PLOS ONE

Journal Requirements:

3. Thank you for stating the following financial disclosure: "This work was supported by the National Social Sciences Foundation of China (Project No. 18CSH019), the Social Sciences Foundation of Beijing (Project No. 20JCC096), and Beijing Municipal Education Commission (Project No. SM201910011007)." 

Reviewers' comments:

Reviewer's Responses to Questions

**Comments to the Author**

1. Is the manuscript technically sound, and do the data support the conclusions?

Reviewer #1: No

Reviewer #2: Partly

2. Has the statistical analysis been performed appropriately and rigorously? 

Reviewer #1: Yes

Reviewer #2: Yes

3. Have the authors made all data underlying the findings in their manuscript fully available?

Reviewer #1: No

Reviewer #2: Yes

4. Is the manuscript presented in an intelligible fashion and written in standard English?

Reviewer #1: No

Reviewer #2: Yes

5. Review Comments to the Author

Reviewer #1: Dear authors and editor,

Many thanks for inviting me to review the manuscript titled “How does the online innovation community climate affect the user's value co-creation behavior: the mediating role of motivation”

(# PONE-D-23-29718) submitting to Plos One.

After reading it, I should point out some issues

1.The version of the abstract is not good, it is recommended that the author can add background and value section, based on the structured abstract.

2. In the Keywords, they are just so-so and the authors keep the order from the capital letter "A-Z". Besides, the number of keywords can be reduced to 5-6 and divided them by semicolon.

3. In the Introduction, the materials of the paper should be just update to 2022, and the authors also can look through the recent and important paper on this topic again. It lacks important literature. In the Literature section, we should link to the recent and most literature, specially 2020-2023. Besides, we also pay more attention to the value and contribution of the paper. The authors should add it, use literature to develop the arguments. The literature should be updated, the following can be cited, I also suggest that.

DOI:10.1108/IJIS-12-2022-0237

DOI:10.1016/j.heliyon.2023.e13192

4.In the literature review, the literature can be updated to the year of 2023. Besides, the literature upon the topic “innovative behavior” can cite the following, I suggest that

DOI:10.1007/s12144-020-01120-5

DOI:10.2147/PRBM.S380989

DOI:10.1108/BJM-09-2019-0335

5.In the section of methodology including selecting data, the authors should add more details into the section as much as possible.

6.The authors should make the definitions of the variables clearly in the section of “variables” .

7. The authors should control the number of table and some of them can be integrated.

8. Before the conclusion section, the authors should add discussion section, comparing the findings of the study with previous studies. However, it is missing.

9.The authors should check all the language over the whole manuscript.

10. In terms of references, due to the fact that empricial research requires a large amount of literature support, the current number of references is relatively small, with at least 60 additional articles with high quality added. Additionally, it is important to ensure that the format meets the requirements of Plos One.

Therefore, based on the value and contribution of the present version, I choose “Major Revision”, the authors should address these issues and welcome the revised version in the future.

All the best

Sep 22, 2023

Reviewer #2: Thank you for the opportunity to review the manuscript. I appreciate the effort put into examining the relationship between online innovation community climate, user motivation, and value co-creation behavior. The study's comprehensive approach, especially with such a large sample size from the OnePlus Community, is commendable.

However, after a thorough review, I have some questions and suggestions that I believe could further enhance the quality and relevance of the research:

1. Research Motives and Cohesion: The choice of examining user’s participation behavior and user’s citizenship behavior in the context of an online innovation community raises questions. Given the focus on innovation, it might be more pertinent to explore behaviors more closely associated with innovation and creative collaboration. As someone deeply involved in creativity and innovation research, I am keen to understand how your model might relate to innovation and creativity outcomes. It might be beneficial to consider adding this as a sequential outcome variable following the value co-creation behaviors you have tested.

2. Theory of Value Co-Creation: While I am not entirely familiar with the theory of value co-creation you employed, it seems to have a broad application in various business dynamics. I would appreciate it if you could elucidate how this theory specifically relates to the context of an online innovation community. This would provide clarity and strengthen the theoretical foundation of your study.

3. Choice of User’s Participation Behavior: The decision to use user’s participation behavior over creative engagement is intriguing. I would recommend citing relevant references to justify this choice and provide a clearer rationale for the same.

In conclusion, I believe that addressing these questions and suggestions can provide a more cohesive and contextually relevant narrative to your research. I hope my feedback is constructive and aids in refining your manuscript further.

6. PLOS authors have the option to publish the peer review history of their article (what does this mean?). If published, this will include your full peer review and any attached files.

Reviewer #1: **Yes: **Assoc. Prof. Dr. Bei Lyu

Reviewer #2: No

---

## [Author Response · Author response to Decision Letter 0]

22 Nov 2023

To Reviewer 1:

1.The version of the abstract is not good, it is recommended that the author can add background and value section, based on the structured abstract.

Reply: 

Thank you very much for your guidance on the Abstract. We have added the background information of the OIC and the value section. The revised abstract is as follow:

Online Innovation Community (OIC) serves as a virtual space for users to exchange products and services, and share knowledge and information. Previous studies have indicated that community climate is an important factor affecting users' value co-creation behavior, however, the influencing process has not been clearly revealed from the perspective of motivation. In this study, we explored the relationship between online innovation community climate (supportive climate and controlling climate), user motivation and value co-creation behavior (user’s participation behavior and user’s citizenship behavior) based on the SOR model. This study sample included 29,835 pieces of information from 3,315 users in 14 product sections of the OnePlus Community which were analyzed with Mplus8.1. The findings revealed that: (1) The supportive climate had a positive impact on user’s citizenship behavior(β=0.042), while the controlling climate exerted a significant positive impact on user’s citizenship behavior (β=0.078) and user’s participation behavior(β=0.099); (2) The need for achievement played a suppressing effect between community climate and user’s participation behavior, the need for power played a suppressing effect between supportive climate and user’s value co-creation behavior, and the need for affiliation played a mediating role between supportive climate and user’s citizenship behavior (β=0.010) and user’s participation behavior(β=0.006); (3) Community trust positively moderated the relationship between the need for achievement and user’s participation behavior(β=0.058) as well as between the need for power and user’s participation behavior(β=0.043).

2. In the Keywords, they are just so-so and the authors keep the order from the capital letter "A-Z". Besides, the number of keywords can be reduced to 5-6 and divided them by semicolon.

Reply: 

We have changed the order by "A-Z" and revised the number of keywords are 5 as you suggested. The revised Keywords are as follow:

Keywords: community climate; motivation; online community; trust; value co-creation

3. In the Introduction, the materials of the paper should be just update to 2022, and the authors also can look through the recent and important paper on this topic again. It lacks important literature. In the Literature section, we should link to the recent and most literature, specially 2020-2023. Besides, we also pay more attention to the value and contribution of the paper. The authors should add it, use literature to develop the arguments. The literature should be updated, the following can be cited, I also suggest that.

DOI:10.1108/IJIS-12-2022-0237

DOI: 10.1016/j.heliyon.2023.e13192

Reply:

There are several recently published papers we have read, some of which published in high impact journals. We have cited these related literatures in line36, line63 and line73.

4.In the literature review, the literature can be updated to the year of 2023. Besides, the literature upon the topic “innovative behavior” can cite the following, I suggest that

DOI:10.1007/s12144-020-01120-5

DOI:10.2147/PRBM.S380989

DOI:10.1108/BJM-09-2019-0335

Reply: 

Thank you very much for the suggestion about the literature review, we have cited these important literatures in line97, line114, Line128, line147, line193 and line224.

5.In the section of methodology including selecting data, the authors should add more details into the section as much as possible.

Reply: 

We have added some information about the OnePlus mobile community and the operational measures. The revised contents were:

This study analyzed the OnePlus Community (https://www.oneplusbbs.com), established in 2013 by OnePlus mobile phone officials. It is a virtual community where users can exchange products and services, as well as share knowledge and information. In OnePlus Community, users have opportunities to express their views and share their experiences. According to the report by Counterpoint in the first half of 2023, OnePlus was recognized as one of the top ten brands with the fastest growth in global market share and it had more than 15 million community users in 196 countries around the world. The users in OnePlus Community are highly active, the rich user resources owned by the community contribute to this research work. 

In order to collect user data, members of the research group registered as community users and gathered information from OnePlus Community Forum, including the number of new posts, the number of comments, and the number of posts selected as "hot". As required by the community, users can only discuss product experience and improvement suggestions in the corresponding product section. To collect users’ product experience properly, we sampled 14 product sections in the community by Python (as of data collection, a total of 17 sections, excluding three sections beyond the research purpose, including Youleyuan, Qingsheying and Official activities) as the primary source for information collection. Since only the latest 1,000 pages of posts and related information (19 posts per page, 266,000 posts in total) from the time of data capture were available on the website of the target community, we were unable to obtain all the posts of the community since its establishment. Still, the crawled data were sufficient to meet our research needs, and the data posted up to October 4, 2021 were collected.

The data required for the research involves both users and products, which were collected in three rounds: in the first round, posts that were officially rated as “hot” in the community were collected, as shown in Figure 2, including the popularity value, product section where the post was located, and the link to the personal website. A total of 38,894 posts were collected. In the second round, leveraging the uniqueness of each user's personal website has been obtained in the first round, basic user information was collected, including user's name, ID, number of cheers, displayed items of user information, online time, number of friends, etc. In the third round, the number of comments deleted by the authority, the number of comments on the post, and the number of views on the post collected in the first round were collected. Finally, a total of 676,380 pieces of information were collected.

The inclusion and exclusion criteria are as follows: (1) official activity posts; (2) posts and user information with incomplete information; and (3) only one post randomly included for the same user. As the collection of user information data in this study was related to the posts by users, if all posts by the same user were included, the same user information would be collected repeatedly for users who post multiple times, which would cause problems for data cleaning and sorting. In addition, we randomly deleted multiple posts from the same user to ensure the accuracy of the results. Totally 29,835 pieces of information from 3,315 users were finally included for analysis.

6.The authors should make the definitions of the variables clearly in the section of “variables”.

Reply:

 We have added the definitions of the variables, the revised contents were as follows:

(1) Dependent variable: User participation behavior (UPB) refers to the behavior that users actively participate in value-creating activities in the community, so it was measured by the number of posts by users[12]； user citizenship behavior (UCB) refers to the behavior that users spontaneously safeguard users' interests and create additional value in the community, which is measured by the number of comments posted by users [12].

(2) Independent variable: Supportive climate (SC) refers to the climate in which users are encouraged to post freely, communicate on an equal footing and develop friendly relations in the community [16], so it was measured by the hot value of posts marked by the community. Controlling climate (CC) refers to the flow of information in the community, the words, deeds and behaviors of members were restricted [16], so it was measured by the number of comments on posts that were officially deleted.

(3) Mediating variable: The need for achievement (NA) refers to the need of users to achieve goals and pursue excellence based on their own interests and passions [43], so it was represented by the number of cheers users get in the OnePlus Community [56]. The need for power (NP) refers to the user’s need to influence or control others by acquiring social status [56], so it was indicated by the different user group levels that users belong to in the community. There are 11 user groups in the OnePlus Community, and different user group levels are set according to the behaviors of users participating in community activities. Users with stronger community power have a wider scope of activities and comment permissions than users with weaker power. The higher the user group level is, the stronger the influence on others is. The need for affiliation (NFA) refers to the need of users to pursue friendliness and value interpersonal relationships [56], so it was measured by the number of friends a user has in the community.

(4) Moderating variable: Community trust (TR) refers to the user’s trust in the community [52]. In this study, we argue that users think they belong to a certain group. The higher the identification with the community is, the more they believe that the community can represent their own image, the more they trust the community, the more likely they are to display their personal information in the community. Therefore, community trust was measured by items in which users display personal information in the community.

(5) Control variable: User age in community (UAC): UAC can have a significant impact on users’ participation in community activities [58]. In this study, it was represented by the time between user registration and data collection. Product section in community (PSC): the OnePlus Community has set up multiple product sections according to product types and stipulates that users must post content under the corresponding section; otherwise, the post will be deleted. For users, they often have rich experience in using the products they purchased, so they are more likely to share their experience in a fixed product section instead of participating in the activities of other product sections. For this reason, we used PSC as a control variable, which was indicated by the product section to which the user’s content belongs.

7. The authors should control the number of table and some of them can be integrated.

Reply: 

We have integrated the table of the moderating role of the community trust between the need for achievement and the user’s participation behavior, the need for power and the user’s participation behavior, to make the results more clearly. The revised contents were:

We employed hierarchical regression to test the moderating effect of community trust. Since the need for achievement did not have a significant impact on user’s citizenship behavior, only the mediating role of community trust between the need for achievement and user’s participation behavior, and the need for power and affiliation on user’s value co-creation was analyzed. To avoid collinearity, the variables of the need for power and community trust were centralized before testing the mediating effect, and then the interaction item was calculated. Given the length constraints of the article, the valid results of the hypotheses were presented in Table6.

Table 6: The test results of the moderating effect

Variable Participation behavior Participation behavior

 Model 1 Model 2 Model 3 Model 4 Model 5 Model 6

UAC 0.220*** 0.260*** 0.264*** 0.220*** 0.011 0.012

PSC -0.015 0.015 0.024 -0.015 -0.027 -0.027

NA -0.128*** -0.142*** 

NP 0.413*** 0.417***

TR 0.143*** 0.125*** 0.062*** 0.056**

Interaction item 0.058** 0.044**

F 94.066 39.087 9.355 94.066 267.109 7.510

R2 0.054 0.076 0.078 0.054 0.185 0.186

ΔR2 0.054 0.022 0.003 0.054 0.131 0.002

In Model 3 of Table 6, it can be seen that the interaction term between the need for achievement and community trust had a positive impact on user participation behavior (β = 0.058, p < 0.01), and H6a was validated. Additionally, the interaction term between the need for power and community trust exhibited a positive impact on user participation behavior (β = 0.044, p < 0.01), and H6c was validated. However, other hypotheses were not validated.

8. Before the conclusion section, the authors should add discussion section, comparing the findings of the study with previous studies. However, it is missing.

Reply: 

We are honored to receive the comments and guidance about the discussion, we integrated the conclusion and discussion to make the results more clarify. The revised discussion was:

This study explored the impact of the online community climate on users’ value co-creation behavior in the OnePlus Community and analyzed the mediating role of motivation and the moderating role of community trust. The findings are as follows: 

(1) Supportive climate positively affected user’s citizenship behavior, and controlling climate positively affected user’s participation behavior and user’s citizenship behavior. The result is in line with the findings of Zhao et al. [6], whose study showed that supportive climate can promote users' value co-creation behavior by enhancing their psychological capital. Zhu’s (2020) [33] study showed that controlling climate can promote users’ innovation behavior. The stronger the supportive and controlling climate provided by the community, the more it can promote users' information exchange and resource acquisition behavior in the community, thus promoting their value co-creation behavior in the community.

(2) The need for achievement played a negative mediating role between supportive climate and user’s participation behavior, the need for achievement played a suppressing effect between controlling climate and user’s participation behavior; the need for power played a negative mediating role in supportive climate and user’s participation behavior, the need for power played a suppressing effect in supportive climate and user’s citizenship behavior; and the need for affiliation played an mediating role between supportive climate and user’s value co-creation behavior, and similar findings were found in Wann-Yih & Sukoco’s [52] study. These results showed that these three intrinsic motivations can promote the user’s knowledge sharing behavior in OIC.

(3) Community trust had a positive moderating effect between the need for achievement and user’s citizenship behavior, as well as between the need for power and user’s citizenship behavior. The result is in line with the findings of Zhao and Detlor’s (2021) [57] study, indicating that trust plays a moderating effect between the need for power and user’s sharing intensions. This shows that the higher level of community trust strengthens the influence of users' motivation of achievement needs and power needs on their citizenship behavior.

9.The authors should check all the language over the whole manuscript.

Reply: 

We have carefully checked and improved the English writing in the revised manuscript according to the reviewers' comments. And the manuscript has been revised and re-polished by native English speakers. We hope it will meet the standard of PLOS ONE.

10. In terms of references, due to the fact that empirical research requires a large amount of literature support, the current number of references is relatively small, with at least 60 additional articles with high quality added. Additionally, it is important to ensure that the format meets the requirements of Plos One.

Reply: 

Thank you very much for your guidance on the Reference List, some of which published in high impact journals. We have added theses important literatures on the Reference List. The number of the

---

## [Decision Letter · Decision Letter 1]

20 Dec 2023

PONE-D-23-29718R1How does the online innovation community climate affect the user's value co-creation behavior: the mediating role of motivationPLOS ONE

Dear Dr. Gao,

Thank you for submitting your manuscript to PLOS ONE. After careful consideration, we feel that it has merit but does not fully meet PLOS ONE’s publication criteria as it currently stands. Therefore, we invite you to submit a revised version of the manuscript that addresses the points raised during the review process.

This study involved 3315 supporters, as I believe informed consent and ethical approval are necessary.

We look forward to receiving your revised manuscript.

Kind regards,

Chunyu Zhang

Academic Editor

PLOS ONE

Journal Requirements:

Reviewers' comments:

Reviewer's Responses to Questions

**Comments to the Author**

1. If the authors have adequately addressed your comments raised in a previous round of review and you feel that this manuscript is now acceptable for publication, you may indicate that here to bypass the “Comments to the Author” section, enter your conflict of interest statement in the “Confidential to Editor” section, and submit your "Accept" recommendation.

Reviewer #3: All comments have been addressed

Reviewer #4: All comments have been addressed

Reviewer #5: All comments have been addressed

2. Is the manuscript technically sound, and do the data support the conclusions?

Reviewer #3: Yes

Reviewer #4: Partly

Reviewer #5: Partly

3. Has the statistical analysis been performed appropriately and rigorously? 

Reviewer #3: Yes

Reviewer #4: No

Reviewer #5: Yes

4. Have the authors made all data underlying the findings in their manuscript fully available?

Reviewer #3: Yes

Reviewer #4: No

Reviewer #5: Yes

5. Is the manuscript presented in an intelligible fashion and written in standard English?

Reviewer #3: Yes

Reviewer #4: No

Reviewer #5: Yes

6. Review Comments to the Author

Reviewer #3: I am pleased to see that that the authors have taken the previous review comments on board and have set out to deal with every one of them. There has clearly been a great deal of additional work undertaken to address each of the reviewers' concerns, which is duly noted and appreciated. This has made a significant impact on the quality of the paper. Thank you.

Reviewer #4: This paper has tried to study the online innovation community climate affecting the user's value co-creation behavior. This methodology attempted to exploit the SEM using MPlus Program and the multiple regression to test the moderating effect of community trust. However, there are many weak points as follows.

1) This paper aims to analyze the existing data but it does not the development of a theory to create a new model. Although the SOR model has been used for your idea, all constructs in the model are not explained in deep-based theory/concept. It depends on types of the data that you collect such the data. Thus, there is still a lack of newness to design the theory development.

2) The format of the data scale has not been reported to analyze all factors using SEM because the SEM normally will use the same format of the data scale. The paper uses 7 factors, and 1 modulator, but runs correlation 8 which doesn't make sense because correlation analysis will use only 7 factors for SEM analysis.

3) Another test also uses multiple regressions to obtain the results of 15 models using only 5 factors but it does not mention the details e.g. "Why did you select 5 factors?", "How did you use formats of data scale?" etc.

4) All questionnaires used to collect data have not been reported or clarified. Normally, you should select the questions coming from standards that can be referenced from theories, principles, or research articles that have already been published in Journal Q1. This will help solve the problem of the bias questions.

5) Path analysis and Hypothesis are not in the same direction or related.

6) Interpretation:

The paper should focus on the overall work "How to get the objective?", "What is the main important point?", "Which issues should focus the work?".

The paper should discuss details such as the implications of theory and practice and the contributions and impacts of your work. It also is hard to explain contributions to the fields for rigorous comparison with alternative methods to find the state-of-the-art. In addition, the paper should have more details of the limitations and future work.

Therefore, the above issues will make this research article insufficient in content reliability to guarantee publication in a high-standard journal. The research articles must satisfy the following criteria:

1. The study presents the results of original research.

[Not adequate]

2. Results reported have not been published elsewhere.

-

3. Experiments, statistics, and other analyses are performed to a high technical standard and are described in sufficient detail.

[Not adequate]

4. Conclusions are presented in an appropriate fashion and are supported by the data.

[Not adequate]

5. The article is presented in an intelligible fashion and is written in standard English.

[Should improve]

6. The research meets all applicable standards for the ethics of experimentation and research integrity.

[Not]

7. The article adheres to appropriate reporting guidelines and community standards for data availability.

[Not]

Reviewer #5: The author addressed all the suggestions of both the reviewers. Now the MS is in a shape that can be published.

7. PLOS authors have the option to publish the peer review history of their article (what does this mean?). If published, this will include your full peer review and any attached files.

Reviewer #3: No

Reviewer #4: No

Reviewer #5: No

---

## [Author Response · Author response to Decision Letter 1]

27 Jan 2024

1) This paper aims to analyze the existing data but it does not the development of a theory to create a new model. Although the SOR model has been used for your idea, all constructs in the model are not explained in deep-based theory/concept. It depends on types of the data that you collect such the data. Thus, there is still a lack of newness to design the theory development.

Reply：

Thank you sincerely for your valuable insights and feedback regarding the utilization of the SOR model in our paper. In our study, our objective was not to construct an entirely original theory but rather to conduct an in-depth analysis from an integrated perspective. Mehrabian and Russell (1974) extended the SOR model based on environmental psychology, emphasizing that external stimuli (S) influence an individual's cognition, mood, and state (O), subsequently shaping their attitude and behavioral response (R). Drawing upon the SOR framework, we aimed to explore the relationship between online innovation community climate, user motivation and value co-creation behavior.

In this paper, the stimulus factor (S) encapsulates environmental elements, serving as a catalyst for users' perceptual and behavioral changes. The supportive and controlling climates within online communities are integral components of these stimulus factors, directly or indirectly impacting individual responses. The organism factor (O) encapsulates the cognitive and emotional states of individuals, acting as a mediator in the interaction between external stimuli and users' ultimate responses. Reaction (R) is expressed in the form of users' attitude or behavior, and the psychological reaction of users in online communities will ultimately affect their behavior in online communities. Therefore, we put forward the theoretical framework of this paper based on SOR model.

Our research methodology was not a purely data-driven approach; instead, it is grounded in the central question: 'What type of online community climate fosters optimal user participation in value co-creation?' In pursuit of this inquiry, we transposed the Stimuli, Organism, and Response (SOR) theoretical framework, integrating classical motivation theories. This synthesis aims to systematically elucidate the procedural mechanisms underlying the impact of community-supportive climate and controlling climate on users' participation behavior and citizenship behavior, respectively.

This study contributes to the expansion of the application scope of the SOR theoretical framework, enhancing its relevance within the dynamic landscape of online communities. Furthermore, it enriches the application of the SOR theory within the Internet milieu, offering a reinterpretation of the nuanced connotations associated with "S", "O" and "R" in this contemporary context.

We acknowledge the importance of constructing original theories and wish to assure you that it is a direction we are actively pursuing. We greatly appreciate your constructive guidance, and we are committed to enhancing the depth and clarity of our theoretical framework as we continue our research journey.

2) The format of the data scale has not been reported to analyze all factors using SEM because the SEM normally will use the same format of the data scale. The paper uses 7 factors, and 1 modulator, but runs correlation 8 which doesn't make sense because correlation analysis will use only 7 factors for SEM analysis.

Reply: 

Thank you sincerely for your insightful observation regarding the number of factors in correlation analysis. As you correctly pointed out, our paper includes 8 factors in the correlation analysis, encompassing 7 primary factors and 1 modulator. We appreciate the opportunity to clarify the rationale behind this approach.

The additional factor introduced in the correlation analysis is rooted in our hypothesis about the potential correlation between the online community climate and community trust. Specifically, we think that a high level of supportive climate within the online community may contribute to an increased level of trust among users. Consequently, we deemed it essential to explore and analyze the correlation between various dimensions of the online community climate, such as supportive climate and controlling climate, and the trust users place in the community.

Furthermore, we draw inspiration from the work of other scholars, for example, Zhang and Bartol’s (Linking empowering leadership and employee creativity: The influence of psychological empowerment, intrinsic motivation, and creative process engagement. The Academy of Management Journal. 2010) study took a similar analytical approach. In their research, the correlation between independent variables and moderating variables was explored, providing a precedent for our analytical methodology.

3) Another test also uses multiple regressions to obtain the results of 15 models using only 5 factors but it does not mention the details e.g. "Why did you select 5 factors?", "How did you use formats of data scale?" etc.

Reply：

Thank you for your insightful observation regarding the testing approach in our paper. In our study, we aimed to explore the moderating effect of community trust on the relationship between user motivation and user value co-creation behavior. User motivation was categorized into three dimensions: Need for Achievement, Need for Affiliation, and Need for Power. Similarly, user value co-creation behavior comprised user participation behavior and user citizenship behavior. Consequently, we anticipated six paths when testing the moderating effect of community trust. However, upon empirical testing, we identified that the path between the need for achievement and user citizenship behavior is not significant. We also draw inspiration from the work of other Journal Q1. In light of this, we focused our analysis on the remaining five paths where significant moderating effects were observed. This adjustment is explicitly mentioned in the paper from lines #478 to #480. 

Community trust in our study is defined as users' trust in the community, manifested by their identification with the community. The stronger their identification with the community, the more they believe the community represents their own image, leading to increased trust. To measure community trust, we utilized items related to users' willingness to display personal information within the community.

4) All questionnaires used to collect data have not been reported or clarified. Normally, you should select the questions coming from standards that can be referenced from theories, principles, or research articles that have already been published in Journal Q1. This will help solve the problem of the bias questions.

Reply： 

We deeply appreciate your valuable suggestion regarding the data collection process. Your insights on the importance of selecting questions from established standards referenced in theories, principles, or previously published research articles in Journal Q1 are duly noted.

In this paper, the data collection primarily relies on objective data crawled by Python, not questionnaires to measure the variables. In order to collect user data, members of the research group registered as community users and gathered information from the OnePlus community forum. To collect users’ product experience properly, we sampled 14 product sections in the community by Python (as of data collection, a total of 17 sections, excluding three sections beyond the research purpose, including Youleyuan, Qingsheying and Official activities) as the primary source for information collection. 

The data required for the research involves both users and products, which were collected in three rounds: in the first round, posts that were officially rated as “hot” in the community were collected, including the popularity value, product section where the post was located, and the link to the personal website. A total of 38,894 posts were collected. In the second round, leveraging the uniqueness of each user's personal website has been obtained in the first round, basic user information was collected, including user's name, ID, number of cheers, displayed items of user information, online time, number of friends, etc. In the third round, the number of comments deleted by the authority, the number of comments on the post, and the number of views on the post collected in the first round were collected. Finally, a total of 676,380 pieces of information were collected.

The inclusion and exclusion criteria are as follows: (1) official activity posts; (2) posts and user information with incomplete information; and (3) only one post randomly included for the same user. As the collection of user information data in this study was related to the posts by users, if all posts by the same user were included, the same user information would be collected repeatedly for users who post multiple times, which would cause problems for data cleaning and sorting. In addition, we randomly deleted multiple posts from the same user to ensure the accuracy of the results. Totally 29,835 pieces of information from 3,315 users were finally included for analysis.

The definitions and measurements of these variables are carefully crafted in alignment with established scholarly definitions and the specific context of the OnePlus community. Furthermore, we have diligently marked the reference scholars in Table 1.

Your feedback has been instrumental in refining our approach, and we are grateful for the guidance provided to enhance the rigor and validity of our research.

5) Path analysis and Hypothesis are not in the same direction or related. 

Reply：

Thank you very much for your attention to our research and your feedback on the alignment of path analysis and hypotheses.

We explicitly presented all research hypotheses in Section3, "Research Hypotheses." Following descriptive statistical analysis and correlation analysis, we utilized Mplus8.1 software to test the hypotheses between variables, and we are pleased to report that all indicators fell within the threshold standards. The purpose of the path analysis was to illustrate the relationship between online community climate (supporting climate and controlling climate) and user value co-creation behavior (user’s participation behavior and user’s citizenship behavior). Additionally, we employed the Bootstrap method to test the mediating effect of user motivation. However, we found that some results are inconsistent with the hypothesis.

The findings revealed that the need for achievement exerts a suppressing effect between community climate and user’s participation behavior, the need for power exerts a suppressing effect between supportive climate and user’s value co-creation behavior, and the need for affiliation plays a mediating role between supportive climate and user’s citizenship behavior and user’s participation behavior. We also explained the reasons for the inconsistent results in Page 22, #451-475 in the paper. Finally, hierarchical regression analysis was employed to explore the moderating effect of community trust. The results indicate that community trust positively moderates the relationship between the need for achievement and user’s participation behavior, as well as between the need for power and user’s participation behavior.

We appreciate your valuable feedback and will continue to strive for clarity and comprehensibility in our research. If you have any further suggestions or questions, we would be more than willing to hear them. Thank you very much.

6) Interpretation:

The paper should focus on the overall work "How to get the objective?", "What is the main important point?", "Which issues should focus the work?".

The paper should discuss details such as the implications of theory and practice and the contributions and impacts of your work. It also is hard to explain contributions to the fields for rigorous comparison with alternative methods to find the state-of-the-art. In addition, the paper should have more details of the limitations and future work.

Reply：

We genuinely appreciate your thoughtful insights on our paper, specifically regarding the need for a more focused approach in the implications of theory and practice and the contributions and the limitations.

In response to your constructive feedback, we have diligently incorporated additional details in the sections related to the implications of theory and practice, the contributions of our work as well as the limitations. We trust that these enhancements align with the high standards set by PLOS ONE.

The revised contents are: 

Conclusion and Discussion

How to promote users' participation in value co-creation in online communities is an increasingly important topic. Based on the SOR model, this study sample included 29,835 pieces of information from 3,315 users in 14 product sections of the OnePlus Community which were analyzed with Mplus8.1, explore the impact of the online community climate on users’ value co-creation behavior in the OnePlus Community and analyzed the mediating role of motivation and the moderating role of community trust. The findings are as follows:

(1) Supportive climate positively affected user’s citizenship behavior, and controlling climate positively affected user’s participation behavior and user’s citizenship behavior. The result is in line with the findings of Zhao et al. [6], whose study showed that supportive climate can promote users' value co-creation behavior by enhancing their psychological capital. Zhu’s (2020) [33] study showed that controlling climate can promote users’ innovation behavior. The stronger the supportive and controlling climate provided by the community, the more it can promote users' information exchange and resource acquisition behavior in the community, thus promoting their value co-creation behavior in the community.

(2) The need for achievement played a negative mediating role between supportive climate and user’s participation behavior, the need for achievement played a suppressing effect between controlling climate and user’s participation behavior; the need for power played a negative mediating role in supportive climate and user’s participation behavior, the need for power played a suppressing effect in supportive climate and user’s citizenship behavior; and the need for affiliation played an mediating role between supportive climate and user’s value co-creation behavior, and similar findings were found in Wann-Yih & Sukoco’s [52] study. These results showed that these three intrinsic motivations can promote the user’s knowledge sharing behavior in OIC.

(3) Community trust had a positive moderating effect between the need for achievement and user’s citizenship behavior, as well as between the need for power and user’s citizenship behavior. The result is in line with the findings of Zhao and Detlor’s (2021) [57] study, indicating that trust plays a moderating effect between the need for power and user’s sharing intensions. This shows that the higher level of community trust strengthens the influence of users' motivation of achievement needs and power needs on their citizenship behavior.

Theoretical Contribution

The main theoretical contributions of this study are as follows: 

（1）This paper reveals the influence mechanism of online community climate on users’ value co-creation behavior, and expands the related research on user behavior in online community. Earlier studies have shown that the online community climate can have an impact on users’ value co-creation behavior [6], but there is a lack of detailed research on the relationship between the two. In this study, we categorized community climate into supporting climate and controlling climate, users’ value co-creation behavior into user’s citizenship behavior and user’s participation behavior, then we explored the impact of online community climate on these two dimensions, thus promoting the research on ubiquitous value co-creation behavior. 

（2）There is a huge body of literature on online community climate and user’s value co-creation behavior [6, 27], but the mechanism whereby the two interact with each other remains unclear. It is found that achievement need motivation plays a suppressing eff

---

## [Editor Report · Decision Letter 2]

6 Feb 2024

PONE-D-23-29718R2How does the online innovation community climate affect the user's value co-creation behavior: the mediating role of motivationPLOS ONE

Dear Dr. Gao,

Thank you for submitting your manuscript to PLOS ONE. After careful consideration, we feel that it has merit but does not fully meet PLOS ONE’s publication criteria as it currently stands. Therefore, we invite you to submit a revised version of the manuscript that addresses the points raised during the review process.

Please provide the approval form of the Ethics Review Committee.

We look forward to receiving your revised manuscript.

Kind regards,

Chunyu Zhang

Academic Editor

PLOS ONE
---

## [Editor Report · Decision Letter 3]

14 Mar 2024

How does the online innovation community climate affect the user's value co-creation behavior: the mediating role of motivation

PONE-D-23-29718R3

Dear Dr. Gao,

We’re pleased to inform you that your manuscript has been judged scientifically suitable for publication and will be formally accepted for publication once it meets all outstanding technical requirements.

Kind regards,

Chunyu Zhang

Academic Editor

PLOS ONE
---

## [Editor Report · Acceptance letter]

8 Apr 2024

PONE-D-23-29718R3 

PLOS ONE

Dear Dr. Gao, 

I'm pleased to inform you that your manuscript has been deemed suitable for publication in PLOS ONE. Congratulations! Your manuscript is now being handed over to our production team.

Kind regards, 

on behalf of

Dr. Chunyu Zhang 

Academic Editor

PLOS ONE